# Modelling the Spatial Distribution of ASF-Positive Wild Boar Carcasses in South Korea Using 2019–2020 National Surveillance Data

**DOI:** 10.3390/ani11051208

**Published:** 2021-04-22

**Authors:** Jun-Sik Lim, Timothée Vergne, Son-Il Pak, Eutteum Kim

**Affiliations:** 1College of Veterinary Medicine and Institute of Veterinary Science, Kangwon National University, Chuncheon 24341, Korea; sgjun3@gmail.com (J.-S.L.); paksi@kangwon.ac.kr (S.-I.P.); 2UMR ENVT-INRAE 1225, Ecole Nationale Vétérinaire de Toulouse, 31300 Toulouse, France; timothee.vergne@envt.fr

**Keywords:** African swine fever, wild boar, surveillance, spatial epidemiology, Korea

## Abstract

**Simple Summary:**

Since African swine fever (ASF) virus in wild boar populations can spill over to domestic pigs, it is crucial to understand the disease determinants in the wild compartment. However, the imperfect detection sensitivity of wild boar surveillance jeopardizes our ability to understand ASF spatial distribution. In this study, we used national surveillance data of ASF in wild boars collected in the Republic of Korea from 2019–2020 to model the spatial distribution of ASF-positive carcasses for two successive study periods associated with different surveillance intensity. The model allowed us to identify disease risk factors in the Republic of Korea, determine the spatial distribution of the risk of ASF, and estimate the sensitivity of surveillance. The outputs of this study are relevant to policy makers for developing and improving risk-based surveillance programs for ASF in wild boars.

**Abstract:**

In September 2019, African swine fever (ASF) was reported in South Korea for the first time. Since then, more than 651 ASF cases in wild boars and 14 farm outbreaks have been notified in the country. Despite the efforts to eradicate ASF among wild boar populations, the number of reported ASF-positive wild boar carcasses have increased recently. The purpose of this study was to characterize the spatial distribution of ASF-positive wild boar carcasses to identify the risk factors associated with the presence and number of ASF-positive wild boar carcasses in the affected areas. Because surveillance efforts have substantially increased in early 2020, we divided the study into two periods (2 October 2019 to 19 January 2020, and 19 January to 28 April 2020) based on the number of reported cases and aggregated the number of reported ASF-positive carcasses into a regular grid of hexagons of 3-km diameter. To account for imperfect detection of positive carcasses, we adjusted spatial zero-inflated Poisson regression models to the number of ASF-positive wild boar carcasses per hexagon. During the first study period, proximity to North Korea was identified as the major risk factor for the presence of African swine fever virus. In addition, there were more positive carcasses reported in affected hexagons with high habitat suitability for wild boars, low heat load index (HLI), and high human density. During the second study period, proximity to an ASF-positive carcass reported during the first period was the only significant risk factor for the presence of ASF-positive carcasses. Additionally, low HLI and elevation were associated with an increased number of ASF-positive carcasses reported in the affected hexagons. Although the proportion of ASF-affected hexagons increased from 0.06 (95% credible interval (CrI): 0.05–0.07) to 0.09 (95% CrI: 0.08–0.10), the probability of reporting at least one positive carcass in ASF-affected hexagons increased from 0.49 (95% CrI: 0.41–0.57) to 0.73 (95% CrI: 0.66–0.81) between the two study periods. These results can be used to further advance risk-based surveillance strategies in the Republic of Korea.

## 1. Introduction

African swine fever (ASF), caused by the African swine fever virus (ASFV), is a highly contagious viral disease that affects both domestic and wild pigs. Symptoms of ASFV infection include fever, hemorrhage, vomiting, and diarrhea, and nearly 100% mortality can occur with some strains, including the ASFV genotype II that is circulating in the Republic of Korea [1]. Owing to the absence of treatment and vaccine, ASF has imposed a significant socioeconomic burden on the pork industry and caused a negative impact on the environment, triggering trophic cascade effects on biodiversity in the wildlife ecosystem [2,3].

Since ASFV can spill over between wild boars (*Sus scrofa*) and domestic pigs, it is crucial to understand the disease dynamics in both species. Therefore, many countries have developed surveillance systems for wild boars [4,5,6,7,8]. Since ASF has a short infectious period due to severe clinical symptoms, which rapidly leads to death [9], contact with ASF-positive carcasses or contaminated environment and consumption of the ASF-positive carcass by their fellows are considered to be one of the main drivers for the transmission of ASF in wild boar populations. Thus, one of the most effective strategies for ASF control among wild boars is the detection and disposal of ASF-positive carcasses and disinfection of the surrounding contaminated environment [10,11,12]. However, in East and Southeast Asia, wild boar surveillance has not been conducted homogeneously across regions, with many countries having only reported outbreaks in farms [13].

In the Republic of Korea, the first case of ASF was reported in a domestic pig farm on 16 September 2019. Although the government culled the ASF-positive farm and its epidemiologically related farms, it seemed that ASF could be epidemic on domestic pig farms. Thus, the government culled pigs from 261 pig farms in the affected counties to mitigate the spread of ASF [14]. By early October, only 14 cases of ASF in domestic farms were reported and the epidemic was considered to be over among domestic pig farms [15]. Meanwhile, the passive wild boar surveillance, highly dependent on citizens reporting carcasses, detected the first ASFV-positive wild boar carcass on 2 October 2019. Later, in November 2019, the government expanded the surveillance system to include hunting, trapping, and active carcass search by field teams funded by the Ministry of Environment, the Forest service, and the Ministry of National Defense [16]. Consequently to this increased surveillance effort, the number of reported cases increased in January 2020; these were mainly from wild boar carcasses.

It is known that the diseased animals behave differently from healthy animals [17,18,19]. For ASF-positive wild boars, it is considered that they prefer the environments that help relieve symptoms and provide shelter, including forest, shrubs, or water-related areas, which provides cool and moist resting sites with sufficient cover and silence. Thus, it can be helpful to analyze the environmental characteristics where ASF-positive wild boars were found for improving risk-based surveillance strategies and optimizing interventions [8,20,21].

However, most wildlife surveillance systems have limited detection sensitivity due to dependency on public reporting, limited resources, and the difficulty of accessing wildlife populations. Due to these reasons, the risk variables identified in logistic regressions, which is the most common statistical method for risk factor analysis, could be associated with disease risk but also with surveillance effectiveness [22,23]. Also, without adjustment for imperfect detection, it is likely that logistic regressions estimate biased estimates for risk factors and result in biased risk mapping [24,25]. Furthermore, it cannot be determined whether this increase in reports was primarily because of increased surveillance sensitivity or increased prevalence of the disease. Thus, it was difficult to evaluate the dynamics of ASF spread among the wild boar population in South Korea. Because of the uncertainty of the risk of ASF transmission, as of August 2020, restocking of pigs into the farms that were stamped out in the affected regions was still prohibited.

One of the approaches to overcome limited detection sensitivity in epidemiological research is the use of zero-inflated count models, which divide the zeros into true and false zeros at the epidemiological unit level; the former represents true negatives, i.e., disease-free units with zero case reports, and the latter indicates false negatives, i.e., affected units with zero case reports. This makes it possible to disentangle risk factors for the presence of the disease and detection sensitivity [23]. Furthermore, the model can be utilized for the estimation of the number of false negative units and therefore of the sensitivity of the surveillance and to clarify the actual risk of a disease by accounting for reporting bias [22,26].

The objectives of this study were to (1) identify the risk factors associated with the presence of ASF and those associated with the reporting rate of ASF-positive carcasses, (2) estimate the prevalence and sensitivity of the surveillance system, and (3) understand the dynamics of ASF infection among the wild boar population. To do so, a spatial zero-inflated Poisson model (SZIP) was adjusted to the temporal patterns of reported ASF-positive wild boar carcasses in the Republic of Korea.

## 2. Materials and Methods

### 2.1. Surveillance Data

The reported data for wild boars were retrieved from the surveillance system database developed by the Ministry of Environment, Republic of Korea. Passive surveillance was performed in wild boars right from the beginning of the epidemic (August 2018); however, it is expected that the effectiveness of this surveillance stream is highly dependent on the search efforts and public awareness. Consequently, in November 2019, as the cases of ASF in wild boars had been continuously reported in South Korea, the government decided to expand the surveillance system into active and passive surveillance: (1) active surveillance included hunting, trapping, and active carcass search by the Ministry of Environment, the Forest service, and the Ministry of National defense; (2) passive surveillance include wild boar carcass reporting by the public with financial incentives. Accordingly, the carcasses were reported by both the active and passive surveillance system. Since the expansion of the wild boar surveillance system, there has been intensive hunting activity in South Korea [16], reaching 12,255 wild boars during January 2020, which was more than twice the number of wild boars per month before the enhanced surveillance system was introduced [27]. As a result of the enhanced surveillance system, a total of 97,045 wild boars were captured or hunted during 2020 [28].

The collected data included the type of report (public reporting or active searching), report date, diagnosis date, species (e.g., wild boar), type of specimen (found-dead carcass, trapped, or hunted wild boar), administrative addresses, and geographic coordinates. All cases were diagnosed as infected with the detection of ASFV with polymerase chain reaction in National Institute of Environmental Research. For the sake of consistency and homogeneity of the data collection process, only the cases derived from carcasses were included in this study.

During the whole study period, the number of reported ASF-positive wild boar carcasses increased substantially after the implementation of the enhanced surveillance protocols in January 2020 (Figure 1). Considering the reporting rates of ASF-positive wild boar carcasses, we decided to divide the study period into two parts as of 19 January 2020 (break point in Figure 1): (1) the first period being from 2 October 2019 to 18 January 2020, and (2) the second period being from 19 January 2020 to 28 April 2020 (Figure 1).

The study region was located in the northern regions of South Korea, bordering North Korea (Figure 2). The region included the Demilitarized Zone (DMZ) along the border and a 10-km wide Civilian Control Zone (CCZ) from the DMZ. It is permitted to manage agricultural activities including cropping and livestock farms in the CCZ but not to live inside it. Moreover, the military facilities were located in the CCZ, and there are some regions where landmines are still present [16].

Considering the imperfect detection in the surveillance of wild boars, the regions that reported ASF-positive carcasses and their neighboring regions were selected as the study sites and were marked using the second-level administrative regional (si-gun-gu) boundaries. These regional boundaries followed the 2019 Administrative District Boundary for the Republic of Korea from Statistics Korea [29]. Since the movement of wild boars does not follow the boundary of the administrative regions and low spatial resolution could limit the spatial analyses, the study region was partitioned into 1237 regular hexagons with a 3-km diameter. For each of the two study periods, as defined by the breakpoint, the number of the ASF-positive carcasses that were reported was counted in each hexagon.

### 2.2. Spatial Zero-Inflated Poisson Regression Model

Zero-inflated count models have been applied in veterinary epidemiological research to identify risk factors of animal diseases while adjusting for imperfect detection [26,30,31]. These models assume the unknown disease status of epidemiological units as a latent variable following a Bernoulli process. The number of reported cases in the disease-affected units is used to adjust a count process defined by a Poisson or a negative binomial distribution. Thus, the models allow inference of the affected epidemiological unit that are associated with zero counts. Moreover, the models can be extended to incorporate covariates as a form of logistic and count regression to identify and disentangle the risk factors associated with the presence of disease (logistic part) or sensitivity of surveillance (count part). In this study, similar to previous studies [26,30], spatial zero-inflated Poisson model (SZIP) was adjusted to the reported ASF-positive wild boar carcass data. The model utilizes Poisson distribution of the number of reports assuming spatial autocorrelation on the risk of the disease. Therefore, we included an intrinsic conditional autoregressive component in the logistic part of the model. SZIP is a mixed model of a spatial logistic and a Poisson regression model. Its associated probability distribution of the counts can be expressed as follows: (1)$$P\left(\;Y=y_{i}\right)\;=\;\left\{\begin{array}{c} \left(1-\pi_{i}\right)+\pi_{i}\;e^{-\lambda_{i}}\;\;\;\;if\;\left(y=0\right)\;\\ \;\;\;\;\;\;\;\;\;\pi_{i}\frac{\lambda_{i}^{y_{i}}e^{-\lambda_{i}}}{y_{i}!}\;\;\;\;\;\;if\;\left(y\geq \;1\right) \end{array}\right.$$
where i is the index for the hexagon; $y_{i}$ is the observed (i.e., reported) number of the ASF-positive carcasses in hexagon i; $\;\pi_{i}$ is the probability of presence of ASF-positive wild boar carcasses in hexagon i; $\lambda_{i}$ is the average number of reported ASF-positive carcasses given the presence of ASF-positive wild boar carcasses for a hexagon i. The model can be extended to account for covariates in the logistic and Poisson parts as follows:(2)$$logit(\pi_{i})=\alpha_{0}+\sum \alpha X_{1}+\omega_{i}+v_{i}$$
(3)$$log(\lambda_{i})=\;\beta_{0}+\sum \beta X_{2}$$
where $\alpha_{0}$ and $\beta_{0}$ are intercepts; α and β are vectors of coefficients; $X_{1}$ and $X_{2}$ are vectors of covariates for logistic and Poisson parts, respectively; $\omega_{i}$ is a spatially structured random effect; and $v_{i}$ is an unstructured random effect. For $\omega_{i}$, the spatial adjacency structure was assumed based on the first-order contiguity between the hexagons [32]. $v_{i}$ was specified based on Gaussian distribution with a mean of zero. Note that $X_{1}$ and $X_{2}$ can be different sets of covariates, which can help to disentangle the factors for the presence and reports of ASF-positive carcasses. The model implicitly divides the zeros into two categories: true zeros and false zeros. The former comes from the absence of reports due to the true absence of ASF-positive carcasses (i.e., a true-negative hexagon). The latter comes from the absence of reports even though ASF was present, which may be due to low surveillance intensity or low number of infected carcasses (i.e., a false-negative hexagon). The set of covariates involved in the Poisson part ($X_{2}$) allows $\lambda_{i}$ (the average number of reports of ASF-positive carcasses in affected hexagons) to vary between hexagons and therefore account for both heterogeneous abundance of positive carcasses and for heterogeneous surveillance efforts. Note that amongst the variables in $X_{2}$, the factors associated with the abundance of positive carcasses cannot be distinguished from the those associated with surveillance effectiveness.

The sensitivity of the surveillance system for each hexagon ($\hat{Sen_{i}}$), which is defined as the probability of detecting at least one ASF-positive carcass in affected hexagons, can be estimated for each hexagon as one minus the probability of no report when ASF-positive carcasses are present ($e^{\hat{-\lambda_{i}}}$), as in $\hat{Sen_{i}}=1-e^{\hat{-\lambda_{i}}}$. Therefore, the probability of being a false negative can be expressed as follows: $\hat{FN_{i}}=1-\hat{Sen_{i}}=e^{\hat{-\lambda_{i}}}$. Note that specificity was assumed to be perfect, i.e. that all ASFV-positive wild boar carcasses were true positives. The probability that at least one case was reported in each hexagon can be estimated by multiplying the risk by the sensitivity of the surveillance system $(\hat{\pi_{i}}\times \hat{Sen_{i}}$). The hexagon-level prevalence $\left(\hat{PrevO}\right)$, i.e. the proportion of affected hexagons, was estimated as follows [30]:(4)$$\hat{PrevO}=\frac{N_{POS}+\sum_{j\in NEG}\hat{\pi_{j}}\;}{Total\;number\;of\;hexagons\;}$$
where, NEG correspond to hexagons with no reported cases, $\underset{j\in NEG}{\sum }\hat{\pi_{j}}$ is estimated number of affected hexagons with no reported cases (i.e., the number of false negative hexagons); $N_{POS}$ is the number of affected hexagons with reports of the ASF-positive carcasses. The hexagon-level sensitivity of the surveillance system ($\hat{SenO})$ was calculated as follows:(5)$$\hat{SenO}=\frac{N_{POS}}{N_{POS}+\sum_{j\in NEG}\hat{\pi_{j}}}$$

### 2.3. Covariates

Considering the symptoms of ASF-infected wild boars, such as fever, dehydration, and hemorrhage, the behaviors of infected wild boars would be associated with those of the feverish animals’ preference for a cool, humid, and aqueous environment. Thus, the covariates were selected based on these characteristics and previous studies [8,21]. Many types of spatial data were unavailable for the border regions with North Korea due to the military restrictions including land cover data. Thus, the variables that were not allowed to be retrieved due to the military restrictions were extracted from remote sensing data.

Because the distribution of ASFV-positive carcasses was expected to be dependent on the distribution of wild boar distributions, we considered as covariates the wild boar habitat suitability, representing the probability that the conditions are suitable to the wild boar. Moreover, to reflect the resting sites for ASF-positive wild boars, enhanced vegetation index (EVI) as an indicator of forestation level and area of rice paddy was used. The covariates to be associated with the symptoms of ASF included area of surface water, presence of wetland, median normalized difference water index (NDWI), median land surface temperature, median rainfall, and median heat load index (HLI). Moreover, the surveillance activity was dependent on human resources, which could make the sensitivity of surveillance heterogeneous by the level of human accessibility. Consequently, we also considered median human density, median elevation, and median slope as putative explanatory variables. Finally, given that more than 50% of ASFV-positive wild boar carcasses were found at less than 10 km from the North Korean border, we also included the distance to North Korea as a putative explanatory variable.

The habitat suitability for wild boars was retrieved from the species distribution model to reflect the distribution of wild boars presented in [33]. The model provided a habitat suitability index ranging from 0 to 1 with a 1-km resolution, which were summed at hexagon level for the analysis. The value indicated the habitat suitability for wild boars in a hexagon as a measure of wild boar distribution. This variable was included in both parts of SZIP, regardless of statistical significance, to adjust for the confounding effect of wild boar distribution on the associations between risk factors and the probability of presence and the number of reports. EVI, representing the level of vegetation activity, was obtained from the Terra MODIS MOD13Q1 Version 6 product, with a geographic resolution of 250 m [34]. Polygons for the rice paddy, surface water, and wetland in the study region were obtained from the Ministry of Environment and utilized to calculate the areas distributed in each hexagon. These polygon data covered the entire area of South Korea. Land surface temperature at day (LSTD) and night (LSTN) were retrieved from the Terra MODIS MOD11A1 Version 6 product [35]. NDWI, which is a measure of liquid water molecules in vegetation canopies ranging from −1 (e.g., area with no water content) to 1 (e.g., area of vegetation with much water), indicates the moisture level of the areas [36,37]. It was calculated using the surface reflectance bands retrieved from the Terra MODIS MOD09A1 Version 6 product, with a 500-m resolution [36,38]. Rainfall data were retrieved from the Automatic Synoptic Observation System, Korea Meteorological Administration [39]. As the data consisted of observed values with their geographic coordinates, the spatial, continuous surfaces of the rainfall in the study regions were estimated using ordinary kriging interpolation. Elevation was obtained from the Space Shuttle Radar Topography Mission 1 Arc-Second Global data version 3.0 [40]. Topographic data including slopes and HLI was calculated using the elevation data. HLI is defined as the estimate of potential annual direct incident solar radiation. Human density was retrieved from the Gridded Population of the World Version 4, Center for International Earth Science Information Network, Columbia University [41], with a geographic resolution of 30 arc-seconds (approximately 1 km). Using the shape file of North Korea downloaded from Global Administrative Areas version 3.6 [42], which represented the administrative map in 2018, the minimal distance from each centroid of a hexagon to North Korea was calculated.

The median of distributions of time-varying variables, such as EVI, LSTN, LSTD, rainfall, and NDWI, were extracted from each source at the highest temporal resolution for the total study period. Consistent variables, including habitat suitability for wild boars, elevation, slope, aspect, area of rice paddy, surface water and wetland, minimal distance to North Korea, HLI, and human density were extracted from each source and equally analyzed for both the study periods. All values of the variables, spatially corresponding to a hexagon, were extracted, and the median of the values was assigned to the hexagons.

Given that wildlife ecosystems have been considered to be nonlinear [43], all covariates were categorized into three levels with a similar number of hexagons at each level to account for nonlinear association between the covariates and the presence or detection of ASF. The area of the wetland was categorized into two levels (none or present) because 834 of the total hexagons had no wetlands.

### 2.4. Model Building

The models were built using the following process: univariable analysis, collinearity analysis, and multivariable analysis, with a forward selection procedure. In the univariable analysis, all variables were tested independently using the Poisson and the logistic parts of the SZIP regression model. In the univariable analysis for both time periods, besides habitat suitability for wild boars, the spatial and nonspatial random terms were included in the logistic part of the regression analysis. In particular, for univariable analysis of the second study period, the spatiotemporal and temporal autocorrelations were adjusted; this was achieved using the binary variable, indicating whether ASF-positive carcasses were reported in the first study period in the hexagon and their neighborhoods (ST_variable), and the count variable for the number of reports during the first study period in the hexagon (T_report) as a confounder in the logistic and Poisson parts, respectively. Variables associated with at least one coefficient for which zero was not included in the 80% credible interval (CrI) of its posterior distribution were included in the multivariable analysis. Collinearity between the variables identified as significant at 20% in univariable analysis was assessed using the Kendall rank correlation test [44]. If the pair of two variables had a correlation >0.7, one of the variables in the pair with further biological implication was included in the univariable analysis. In multivariable analysis, a forward selection procedure was conducted to build the final models based on the deviance information criteria (DIC), which quantified the balance between the complexity and the fit of the model [45]. To satisfy the parsimony principle, the model that was selected for inference was the simplest one with a DIC less than two points greater than that of the model with the smallest DIC (i.e., the simplest with an equal model fit).

All analyses were performed with Bayesian Markov Chain Monte Carlo simulation (MCMC). The MCMC was performed in R software with the “*R2OpenBUGS*” package [46]. The weakly informative prior distributions for the coefficients of the SZIP were set as normal distributions with a mean of 0 and a variance of 10 to enable the information of the observed data to mainly contribute to the posterior distributions. Likewise, for spatial and nonspatial random effects, the weakly informative prior distribution was assumed to follow the inverse Gamma distribution with a mean of 10 and a variance of 100.

Three chains of 100,000 iterations were sampled with a 20,000 burn-in. Convergence was assessed by checking the trace plot visually and using the Gelman–Rubin–Brooks diagnostic [47]. The posterior distribution of the parameters was summarized with the median and the 95% credible intervals (95% CrI). Furthermore, for each hexagon, median values of the risk, sensitivity, the probability that at least one case was reported, and the probability of a false-negative report were plotted as choropleth maps. The discriminatory performance of the models was assessed using the area under the curve (AUC) of the receiver operating characteristic curves for the estimated probability that at least one case was reported in the hexagons. Validations of the two models were conducted for each training and test set. The training sets were selected as the ASF-positive carcasses that were reported the month following the end of each study period. Validation for the model of the first period (2 October 2019–18 January 2020) was conducted with the ASF-positive carcasses that were reported between 19 January 2020 and 18 February 2020 (the 31 positive hexagons in Figure 3B). For the model of the second period (19 January 2020–28 April 2020), the validation was done using the ASF-positive carcasses that were reported between 29 April 2020 and 28 May 2020 (the 29 reported hexagons in Figure 3F).

To compare the hexagon-level prevalence of ASF-positive wild boar carcasses, the hexagon-level sensitivity of the surveillance system and the hexagon-level proportion of false-negative reports between the two study periods were plotted against each of the posterior distributions. A scatterplot for the probability of the presence of ASF-positive wild boar carcasses between the first and second periods was created.

Because of substantial computational demand, the simulations were run with parallel computing using “*parallel*” and “*dclone*” [48] packages in R software [49].

## 3. Results

Among the 569 wild boar cases that were reported between 2 October 2019 and 24 April 2020, 551 cases were from carcasses, 12 cases from trapped wild boars, and seven cases from hunted wild boars. During the first period (from 2 October 2019 to 18 January 2020), the total number of reports of ASF-positive carcasses was 79, distributed among 37 hexagons. For the second period (from 19 January 2020 to 28 April 2020), 472 of the ASF-positive carcasses were reported in 80 hexagons.

The results of univariable analyses for the first and second periods are presented in Table 1 and Appendix A. For the first period, elevation, slope, area of rice paddy and surface water, minimal distance to North Korea, HLI, human population density, EVI, LSTD, rainfall, and NDWI were statistically significant at the 20% level in both the Poisson and logistic parts of the regression analysis; elevation, slope, area of rice paddy, presence of wetland, minimal distance to North Korea, HLI, human population density, EVI, and rainfall were significant at the 20% level only in the Poisson part of the analysis; and the area of rice paddy, minimal distance to North Korea, human population density, LSTD, LSTN, rainfall, and NDWI were significant at the 20% level in only the logistic part of the analysis. For the second study period, elevation, area of rice paddy and surface water, presence of wetland, minimal distance to North Korea, HLI, human population density, EVI, LSTD, LSTN, rainfall, and NDWI were significant at the 20% level in both the Poisson and logistic parts of the regression analysis; elevation, area of rice paddy and surface water, minimal distance to North Korea, HLI, human population density, EVI, LSTD, rainfall, and NDWI were significant at the 20% level in only the Poisson part of the analysis; and the presence of wetland, minimal distance to North Korea, human population density, LSTD, and NDWI were significant at the 20% level in only the logistic part of the analysis. Among the significant variables in the univariable analyses, LSTD was highly correlated with elevation and slope for the first study period (Appendix A); for the second study period, LSTD was highly correlated with slope (Appendix A). As the variables that do not change over time consistently contributed to the risk or the number of reports of ASF cases in affected hexagons, LSTD was excluded from the multivariable analyses.

The results of the SZIP for the first study period are presented in Table 2. The distance from North Korea, wild boar habitat suitability, HLI, and human population density were identified as statistically significant factors for the first period. In the logistic part of the SZIP, the distance from North Korea to the centroid of hexagons was identified as a significant factor for the presence of ASF-positive wild boar carcasses. The estimated odds ratios (OR) were 0.12 (95% CrI: 0.01–0.95) and 0.01 (95% CrI: 0.00–0.30) for the hexagons located between 14 km and 28 km and those located between 28 km and 76 km, respectively, relative to the estimated OR of hexagons located between 0 and 14 km from North Korea. The number of reported cases of ASF-positive carcasses in affected hexagons (Poisson part of the SZIP) was statistically significantly associated with habitat suitability for wild boar, HLI, and human population density. The estimated incidence rate ratios in the hexagons with the second and third levels of habitat suitability for wild boars were 2.33 (95% CrI: 1.02–4.85) and 8.75 (95% CrI: 1.44–55.54), respectively, compared to the first level of habitat suitability for wild boars. For HLI, only the third level of HLI showed a statistically significant positive association (IRR: 0.40, 95% CrI: 0.17–0.93). Although human population density showed a positive association, only the density of 61.26–9448.68 persons per km^2^ was statistically significant (human density of 24.77–61.26 persons per km^2^, IRR: 3.23, 95% CrI: 0.75–17.00; human density of 61.26–9448.68 persons per km^2^, IRR: 4.99, 95% CrI: 1.09–27.88).

ST_variable, T_report, HLI, and habitat suitability for wild boars were identified as significant risk factors for the second period (Table 3). Only the ST_variable, representing whether ASF-positive carcasses were reported in a hexagon and its neighborhood during the first study period, showed a significant positive association with the presence of ASF-positive wild boar carcasses (logistic part of the SZIP), with an OR of 29.90 (95% CrI: 5.92–220.96). T_report, showing the number of reports of ASF-positive carcasses in a hexagon during the first period; habitat suitability for wild boars; elevation; and HLI were identified as statistically significant factors associated with the number of reported carcasses in the affected hexagons (Poisson part of the SZIP). The estimated IRR of T_report was 1.13 (95% CrI: 1.07–1.20). Habitat suitability for wild boars showed a weak positive association (second level of habitat suitability for wild boars: OR: 2.13, 95% CrI: 1.58–2.94; third level of habitat suitability for wild boars: OR: 1.26, 95% CrI: 0.72–2.13). HLI showed a significant positive association (second-level HLI: IRR: 2.92, 95% CrI: 2.03–4.28; third-level HLI: IRR: 3.00, 95% CrI: 2.08–4.44). Although elevation showed a negative association, only the third level of the elevation was significantly significant (second-level elevation: IRR: 0.88, 95% CrI: 0.63–1.24; third-level elevation: IRR: 0.63, 95% CrI: 0.41–0.96).

The assessments of the discriminatory performance of the models were conducted for the training sets (Figure 3A,E for each period) and test sets (Figure 3B,F for each period). The AUCs of the model of the first period were 0.97 (95% CrI: 0.93–0.99) and 0.90 (95% CrI: 0.83–0.94) for the training and test sets, respectively. For the second period, it was estimated as 0.98 (95% CrI: 0.97–0.99) for the training set and 0.95 (95% CrI: 0.93–0.97) for the test set.

With the estimated models, the median values of the risk, sensitivity, and the probability that at least one case was reported were plotted with the number of reports (Figure 3). For the first study period, the hotspots with ASF-positive carcasses were located in the northwest and central north regions (Figure 3C). Sensitivity was lower in the central regions than that in the other regions. Moreover, the regions surrounding those with reported cases had a low surveillance sensitivity (Figure 3D). During the second study period, the maximum value of the probability of the presence of ASF-positive wild boar carcasses in the hexagons was higher than that during the first study period. The hotspot of the central north regions had moved to the southeast (Figure 3G). The overall sensitivity was higher in the second study period than that during the first study period. However, hexagons with relatively low sensitivity were present in the eastern regions (Figure 3H). Figure 4 shows the probability of false negatives in each hexagon during the first and second study periods. During the first study period, there were hexagons with a high probability of false negatives in the central and northwest regions where ASF-positive carcasses were reported. Overall, the probabilities of false negatives during the second period were lower than that during the first period. Nonetheless, there were hexagons with a relatively high probability of false negatives in the central and northeast regions where ASF-positive carcasses were reported.

The hexagon-level prevalence, sensitivity, and proportion of false negative for each period are presented in Table 4 and Figure 5. The number of affected hexagons with no carcasses reported was not statistically different between the two study periods: first pe-riod, 39 (95% CrI: 29–53); and second period, 30 (95% CrI: 20–43). The hexagon-level prev-alence of ASF-positive carcasses was significantly greater during the second period than that during the first (0.09, 95% CrI: 0.08–0.10 vs 0.06, 95% CrI: 0.05–0.07). Hexa-gon-level sensitivity increased from 0.49 (95% CrI: 0.41–0.57) in the first period to 0.73 (95% CrI: 0.66–0.81) in the second period. The proportion of false negative was not statistically dif-ferent between the first (0.03, 95% CrI: 0.02–0.04) and second study periods (0.02, 95% CrI: 0.02–0.03). Figure 6 shows a scatterplot representing a positive relationship (red line in Figure 6) between the probabilities of the presence of ASF-positive carcasses in the first and second study periods.

## 4. Discussion

To control ASF dynamics among wild boar populations, the Korean government has developed and operates surveillance systems. However, wildlife surveillance has limitations of imperfect disease detection because of the dependence on public reporting and limited resources; thus, it is highly likely to create confusion as to whether the identified risk factors are associated with the risk or reporting of ASF-positive carcasses [23], which can also result in uncertainties about the dynamics of ASF among wild boar populations.

In this study, we analyzed the spatial distributions of ASFV-positive wild boar carcasses for two successive periods, while adjusting for the imperfect detection. First, it transpired that between the two periods, the risk factor associated with the presence of ASF-positive carcasses changed from the proximity to North Korea to the proximity of ASF-positive carcasses reported during the previous study period. Second, habitat suitability for wild boars, human population density, HLI, and elevation were associated with the number of reports of ASF-positive carcasses in the affected hexagons; (3) hexagon-level prevalence of ASF-positive carcasses increased between the two periods as well as the hexagon-level sensitivity of ASFV-positive carcass reporting.

For the first study period, our results emphasized that the infection risk of ASFV is significantly increased with decreasing distance from the North Korean border, suggesting that the ASFV could have been introduced from North Korea. This is consistent with the fact that North Korea reported an ASF outbreak on 30 May 2019, before the ASF epidemic in South Korea [50]. South Korea is geographically connected to Asia through North Korea. Although the buffer zone between the two countries, which is also called the Demilitarized Zone, can be a physical barrier to the movement of ASF-infected wild boars, many wildlife species that can fly or swim across rivers flowing between the South and North Korea, including wild boars, can mechanically or biologically spread ASFV from North Korea to South Korea [51,52].

As expected, the results also showed that the habitat suitability for wild boars was associated with the number of reports of ASF-positive carcasses in the affected hexagons. This is likely due to higher densities of wild boars leading to higher contact rates, resulting in a higher incidence and therefore in an increased number of reports of ASF-positive carcasses. Moreover, more ASF-positive carcasses were reported in affected hexagons with low heat load index (HLI). We identified three factors that could explain this: first, wild boars prefer high-moisture and cool-temperature environments [53,54], leading to an increased abundance of wild boars and therefore of ASF-positive carcasses, which is consistent with previous studies [8,21]; second, this observation might be due to the fact that ASFV is more resistant to colder environments, which means that the virus remains active for a longer time, generating more opportunities for transmission [55]; third, it takes more time for wild boar carcasses to decompose in shade than in sunlight [56], which can make detection easier and therefore increase the number of reported carcasses. Usually, human population density is a proxy for human activities and therefore for passive disease detection in veterinary epidemiological research [57,58,59,60]. Consistently, we found that human population density was associated with the number of reported ASF-positive carcasses in the affected hexagons, as has also been identified in other wildlife research [57]. During the first study period, the surveillance was highly dependent on the passive disease detection (i.e., public reporting). In populated regions, there are more accessible—and thus amenable to surveillance—areas due to the more developed traffic systems such as roads and mountain paths. Moreover, the reporters and hunters employed by the government were more likely to get help from local residents to locate carcasses in more densely populated regions. Human population density can be related to the landmine distribution. The area where the landmines were distributed have lower population density, which can lead to lower detection of ASF-positive carcasses. That is, the regions with high human population density had a higher search capacity during the first period.

During the second study period, the fact that the carcasses were already reported during the first period in a hexagon or its neighboring hexagons was a strong risk factor for the presence of ASF-positive carcasses. Unlike during the first study period, the distance to North Korea was not associated with the presence of ASF-positive wild boar carcasses, suggesting that disease distribution was driven by the previous phase of the epidemic rather than by potential reintroductions of ASFV from North Korea.

Moreover, HLI was found to be positively associated with the number of reported carcasses in the affected hexagons, while no such observation was noted during the first study period. This could be due to seasonal effects of HLI. Unlike during the first period, the second period included the spring season. As the regions with high HLI receive more solar radiation, they would have more dense vegetation than the other regions [61,62], and consequently have more moist and cooler microclimates [63]. Thus, these environments can attract more wild boars, which can result in a higher rate of infection and therefore an increased number of reports. Alternatively, it can be interpreted that the infected wild boars prefer this type of environment to hide from hunters or predators. Thus, more ASF-positive carcasses might be reported in such regions. The results also suggest that among the affected hexagons, the higher the altitude of a region, the lower the number of reports. As surveillance was still highly dependent on human resources, it may be difficult to report the carcasses in regions that are difficult for people to access. Furthermore, habitat suitability for wild boars showed a weak statistical association with the number of reports. This may be due to the enhanced surveillance for wild boars during the second study period, which induces changes in the population behaviors. Wild boars flexibly adjust their activity and home ranges to human activities, such as hunting and recreation, to protect themselves from risk [64,65,66]. Owing to the intensive hunting condition in South Korea [16,27,28], wild boars could have left their original home range and shifted to other areas where they could hide from the hunters [65]. This makes the spatial distribution of wild boars unstable and interspersed [64].

The results also suggested that the increased number of reported ASF-positive carcasses from the first to the second study periods was the combined results of the increased prevalence of ASF and an improved sensitivity of the surveillance system. Taken together with the maps for the probability of presence of ASF-positive carcasses and its relationship between the first and second study periods, the probability of false-negative reports, and the increased prevalence for both periods, it can be stated that ASF infection had spatially spread centered on the previously affected regions, especially in the southward direction. Although it is favorable that the overall sensitivity of the surveillance system increased in the second period, it should be noted that some regions remained undetected, and those numbers are considered to be unchanged. These might have contributed to the spatial spread. The movement of the hotspot towards the southeastern direction in the second study period may be explained by the following possibility: there were hexagons with low surveillance sensitivity around the central north hotspot in the first period. Thus, ASFV could have been introduced into these regions and remained undetected during the first study period. The undetected ASF-positive carcasses could have contributed to the infectious events in these regions, which could have generated the hotspot identified in the second study period. Similarly, the undetected hexagons in the second period might contribute to spatial spread in the future: the carcasses undetected in the northeast regions in the second period could contribute to the spread of ASFV in the southern direction because of low surveillance sensitivity in these regions. Furthermore, the “Taebaek mountain ranges” are present around these regions, starting from North Korea and extending to the middle of South Korea. Since ASFV infection among wild boars could follow the boundary of these mountain ranges [67], controlling disease spread may be more demanding due to the high elevation and low human population density.

Considering that the study regions include the border with North Korea, there are areas with unique characteristics: the DMZ (2 km from the border) and CCZ (10 km from the DMZ), which correspond to the reference level of the variable of the distance to North Korea. In these regions, there were no residents because of military reasons. Moreover, there are landmines in these regions. Thus, even if the probability of the presence of the ASF-positive carcasses was high, it would have been difficult to search for such carcasses. Therefore, during the first period, it can be assumed that there were some number of undetected cases in these regions. These undetected carcasses could be a driver of the endemicity of ASF infection among wild boars.

From the standpoint of the fact that the ASF wild boar surveillance systems have limited detection sensitivity, our findings give suggestions for the improvement of ASF wild boar surveillance systems: it is necessary to take actions to deploy more surveillance efforts in the regions that have limited accessibility but are suitable for wild boars, such as less-populated and high-elevation regions in South Korea, and thus have been less surveyed. Moreover, considering that the environmental characteristics of the distribution of ASF-positive carcasses have changed, the surveillance efforts should be guided based on scientific evidence of the seasonal and human-mediated effects on the distribution of wild boars and dynamics of ASF, as well as the geographical proximity of previously reported cases.

In this study, only ASF-positive wild boar found-dead carcasses were included for the statistical analysis. The reasons why we only included ASF-positive found-dead carcasses are that the wild boar is a terrestrial mammal that can move around within its home range. Thus, based on the geographical locations where the wild boars are trapped or hunted, it is difficult to understand the characteristics of the location of ASF-positive wild boars. Moreover, it is highly likely that ASF-positive wild boars are hunted during the incubation period of ASF infection before symptom onset, which indicates that that the environmental characteristics of the geographical locations where the ASF-positive animals were hunted or trapped are not associated with the symptoms of ASFV infection. Furthermore, the main driver of ASF spread among wild boar populations is ASF-positive carcasses. Thus, the surveillance for ASF-positive carcasses can indicate the spatial distribution of ASF and would also reduce the risk of ASF in the population. That is, the detection of wild boar carcasses is the pivotal axis of ASF control strategies in wild boars. Thus, we focused only on ASF-positive carcasses. The impact of not including hunted or trapped animals is expected to be negligible, as these cases account for only 3% of the wild boar cases. Moreover, the sensitivity of surveillance using trapped or hunted wild boars is considered to be lower than that of surveillance using carcasses, suggesting that the sensitivities estimated in this study are the maximum values [8,20,21,68].

This study had two main limitations. First, because of military restrictions, many types of spatial data were unavailable for distances less than 25 km from the North Korean border, which covers 54% of the study region. Consequently, many variables that are potentially related to observed distribution of ASF-positive carcasses, such as road length, density of domestic pig farms, and landmine distribution, could not be included. However, the effects of these variables would be accounted by the human population density, associated with the road density, density of domestic pig farms, and landmine distributions. Second, we did not account for the death time point of the carcasses in the analyses. This might result in bias in the results, since carcasses found during the second study period could be due to the death of a wild boar during the first period. However, the data of time of death was not available, as this estimation is incredibly challenging owing to its dependence on many interrelated factors [56].

## 5. Conclusions

In this study, we analyzed the spatial distribution of ASF-infected wild boars for successive periods, while also accounting for imperfect detection. The epidemiological situation of ASF among wild boar populations seems to have changed into an endemic status, and further, appears to be getting worse. Although the performance of the existing surveillance system has improved, it is possible to improve the sensitivity based on the results of this study to suppress the further spread of ASF. We believe that the factors identified in this study could be useful for developing and improving risk-based surveillance systems.

## Figures and Tables

**Figure 1 animals-11-01208-f001:**
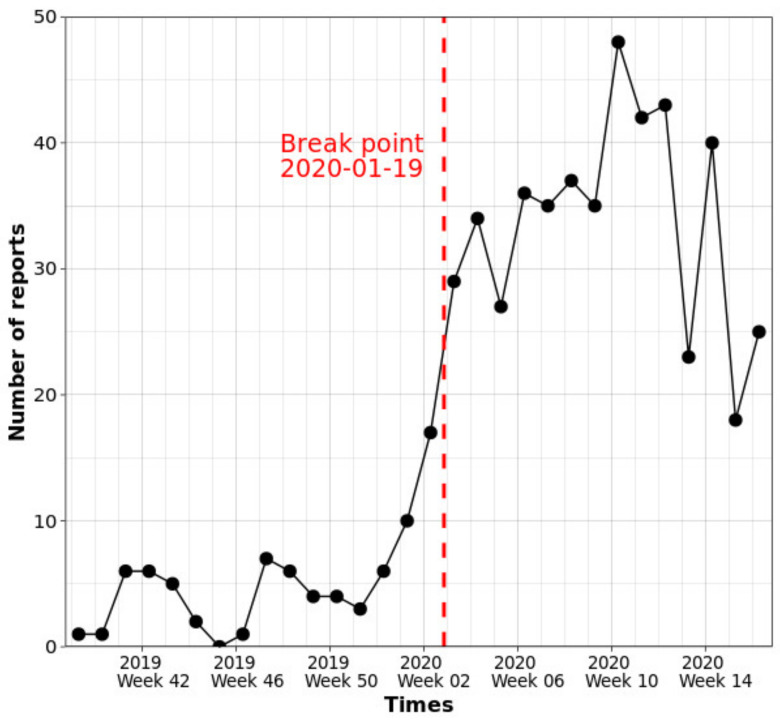
Time series plot for the number of reported ASF-positive wild boar carcasses per week from October 2019 to April 2020: the red dashed vertical line shows the break point on 19 January 2020.

**Figure 2 animals-11-01208-f002:**
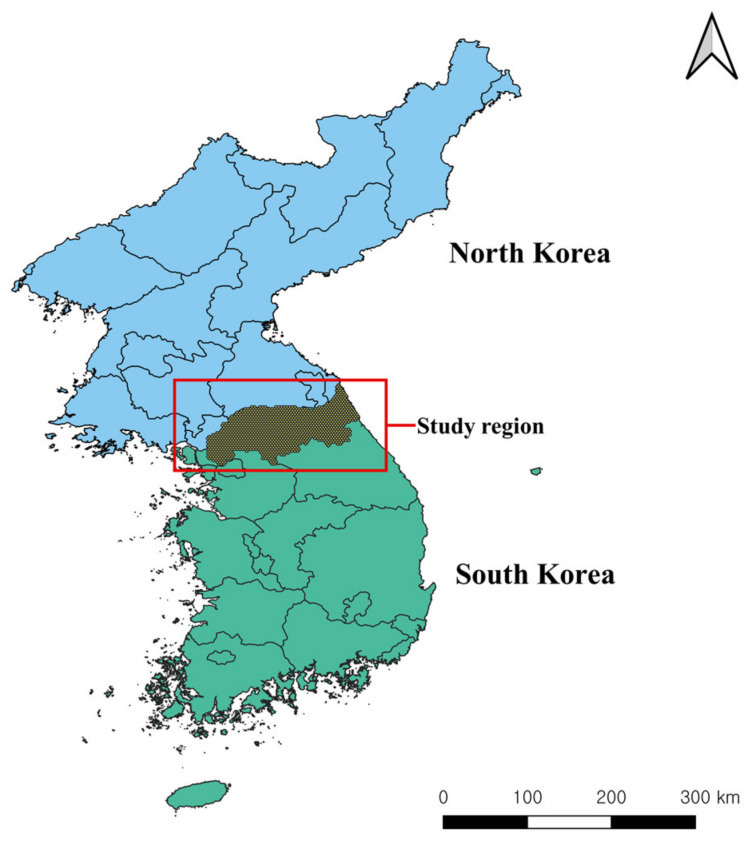
Map showing the location of the study region in South and North Korea. This map was plotted using QGIS version 3.4.11. http://qgis.osgeo.org (accessed on 10 March 2021).

**Figure 3 animals-11-01208-f003:**
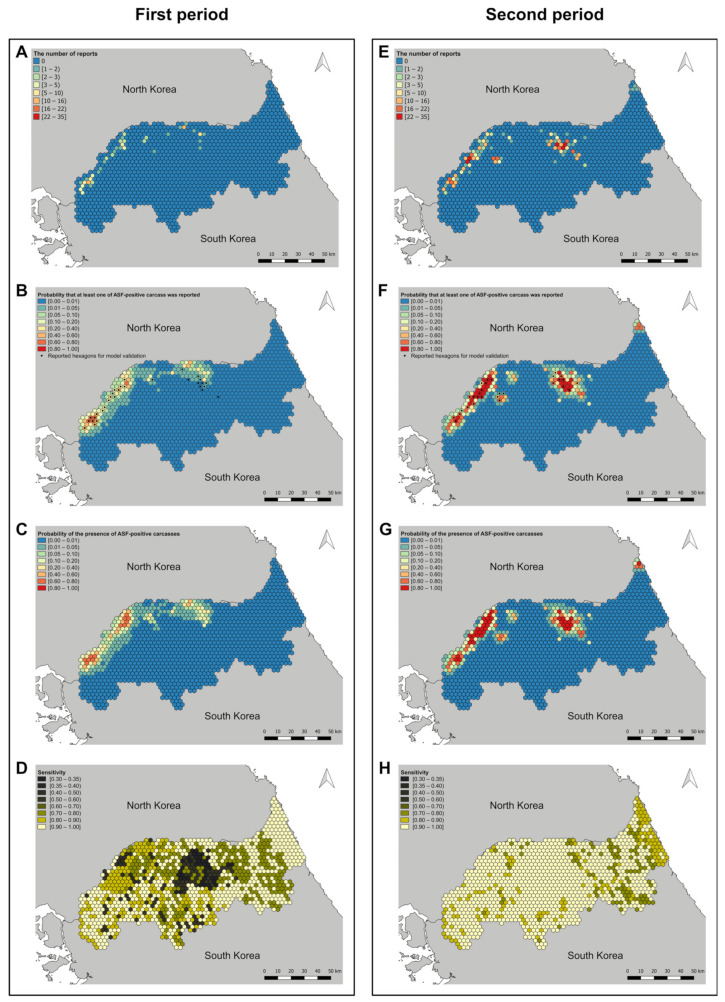
Choropleth maps of the number of reports of African swine fever (ASF) – positive wild boar carcasses ((**A**,**E**) for each period), the median value of the probability that at least one report of ASF-positive wild boar carcass ((**B**,**F**) for each period), the median value of the probability of presence of ASF-positive wild boar carcasses ((**C**,**G**) for each period), and the median value of the sensitivity ((**D**,**H**) for each period) for the first and second periods. These maps were plotted using QGIS version 3.4.11. http://qgis.osgeo.org (accessed on 10 March 2021).

**Figure 4 animals-11-01208-f004:**
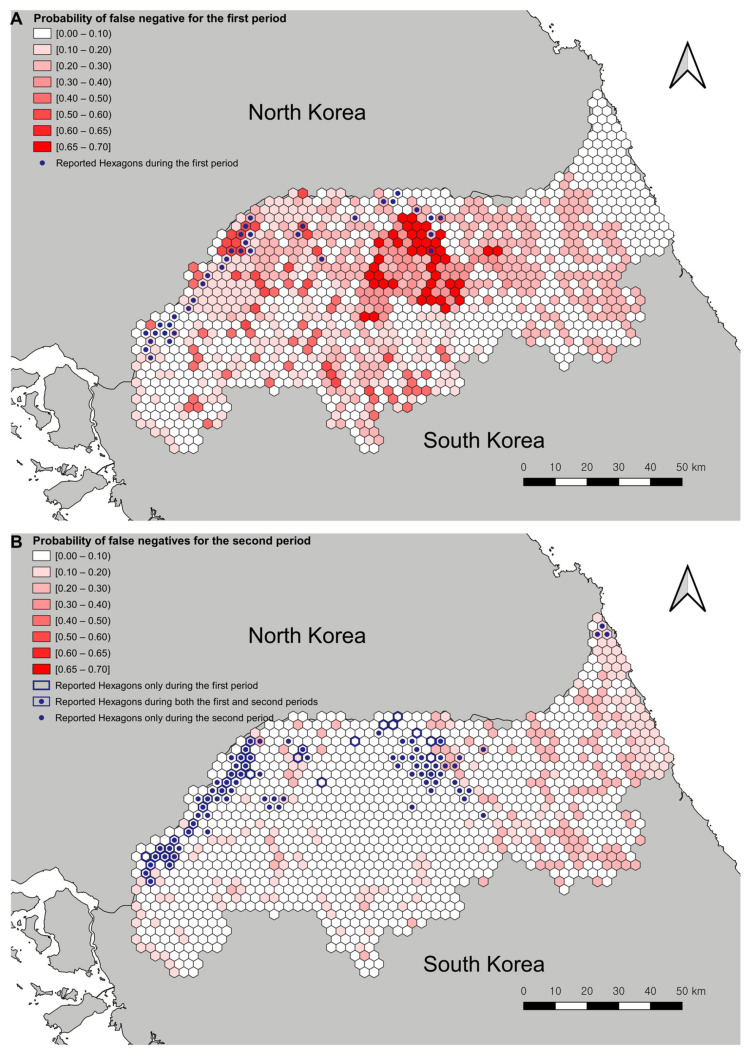
Choropleth map of the probability of false negative during the first (**A**) and second period (**B**). These maps were plotted using QGIS version 3.4.11. http://qgis.osgeo.org (accessed on 10 March 2021).

**Figure 5 animals-11-01208-f005:**
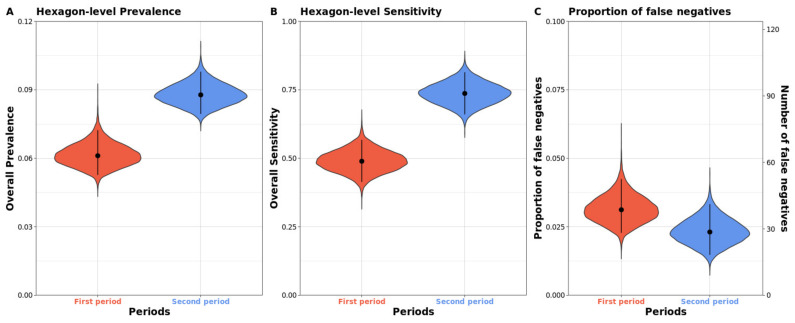
Violin plots for the posterior distribution of the hexagon-level prevalence (**A**), hexagon-level sensitivity (**B**), and proportion and number of the affected but undetected hexagons (**C**) for the first and second periods.

**Figure 6 animals-11-01208-f006:**
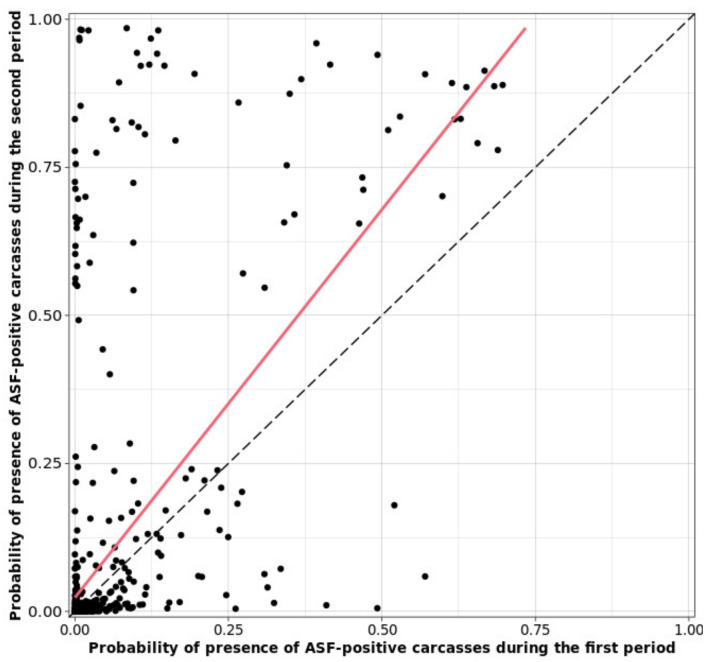
Scatter plot for the relationship between the probability of presence of ASF-positive wild boar carcasses for the first and second periods. Red line indicates the positive relationship between the probabilities of presence of ASF-positive carcasses for the two periods. Black dashed line indicates the diagonal line to help to understand the relationship between the probabilities of the two periods.

**Table 1 animals-11-01208-t001:** Summary of univariable analysis for the first and second periods. A X indicates that the variable had a statistically significant association with the dependent variable in the models, with a significance threshold of 20%.

Variables	First Period (2 October 2019–18 January 2020)				Second Period (19 January 2020–28 April 2020)			
	Tested only in Logistic Part	Tested only in Poisson Part	Tested in Both Poisson and Logistic Parts		Tested only in Logistic Part	Tested only in Poisson Part	Tested in Both Poisson and Logistic Parts	
			Logistic Part	Poisson Part			Logistic Part	Poisson Part
Water ^1^						X		X
Wetland ^2^		X			X		X	
DNorthKor ^3^	X	X		X	X	X	X	X
Elevation		X		X				
Slope		X		X				
HLI ^4^		X	X	X	X		X	X
Area_rice ^5^	X	X	X	X		X		X
Hu_pop ^6^	X	X	X	X	X	X	X	X
LSTD ^7^	X	X		X	X		X	
EVI ^8^		X		X		X		X
NDWI ^9^	X		X	X	X	X	X	X
LSTN ^10^	X					X		X
Rainfall	X	X	X	X	X	X	X	X

^1^ Water: Area of water within a hexagon; ^2.^ Wetland: area of wetland within a hexagon; ^3^ DNorthKor: Distance to North Korea from the centroid of each hexagon; ^4^ HLI: heat load index; ^5^ Area_rice: area of rice paddy within a hexagon; ^6^ Hu_pop: human population density; ^7^ LSTD: land surface temperature at day; ^8^ EVI: enhanced vegetation index; ^9^ NDWI: normalized difference water index; ^10^ LSTN: land surface temperature at night; *: NS: not statistically significant at the 20% level.

**Table 2 animals-11-01208-t002:** Results of the spatial zero-inflated Poisson model for the first period (from 2 October 2019 to 18 January 2020).

Variables	Category	Number of Hexagons	Median of Odds Ratio	95% CrI *	Median of Incidence Rate Ratio	95% CrI *
Habitat suitability for wild boars	(0.29–0.46)	408	Ref.		Ref.	
	(0.46–0.59)	408	0.21	0.02–1.45	2.33	1.02–4.85
	(0.59–0.85)	421	0.07	0.00–1.33	8.75	1.52–56.43
Distance to North Korea (km)	(0–14.62)	408	Ref.		-	
	(14.62–28.16)	408	0.12	0.01–0.95		
	(28.16–76.77)	421	0.01	0.00–0.30		
Heat load index	(0.62–0.81)	408	-		Ref.	
	(0.81–0.84)	408			1.50	0.80–2.98
	(0.84–0.99)	421			0.40	0.17–0.93
Human population density (persons per km^2^)	(15.68–24.77)	408	-		Ref.	
	(24.77–72.36)	408			3.23	0.75–17.00
	(72.36–9448.68)	421			4.99	1.09–27.88

* CrI: Credible interval.

**Table 3 animals-11-01208-t003:** Results of the spatial zero-inflated Poisson model for the second period (from 19 January 2020 to 28 April 2020).

Variables	Category	Number of Hexagons	Median of Odds Ratio	95% CrI *	Median of Incidence Rate Ratio	95% CrI *
Habitat suitability for wild boars	(0.29–0.46)	408	Ref.		Ref.	
	(0.46–0.59)	408	0.75	0.08–5.11	2.13	1.58–2.94
	(0.59–0.85)	421	0.16	0.01–2.72	1.26	0.72–2.13
ST_variable **	No	1100	Ref.		-	
	Yes	137	29.90	5.92–220.96		
T_report ***	Count variable	1237	-		1.13	1.07–1.20
Heat load index	(0.62–0.81)	408	-		Ref.	
	(0.81–0.84)	408			2.92	2.03–4.28
	(0.84–0.99)	421			3.00	2.08–4.44
Elevation (km)	(0–0.20)	408	-		Ref.	
	(0.20 –0.38)	408			0.88	0.63–1.24
	(0.38–1.16)	421			0.63	0.41–0.96

* CrI: Credible interval. ** ST_variable: Binary variable indicating whether the cases were reported in the hexagons and their first-order neighborhoods during the previous period. *** T_report: Count variable indicating the number of the reported cases during the previous period for each hexagon.

**Table 4 animals-11-01208-t004:** The hexagon-level prevalence, sensitivity, and proportion of false negatives for each period.

Value	The First Period (95% CrI *)	The Second Period (95% CrI *)
Total number of hexagons	1237	1237
Number of reported hexagons (True positives)	37	80
Number of undetected hexagons (False negatives)	39 (29–53)	30 (20–43)
Hexagon-level prevalence	0.06 (0.05–0.07)	0.09 (0.08–0.10)
Hexagon-level sensitivity	0.49 (0.41–0.57)	0.73 (0.66–0.81)
Proportion of false negatives	0.03 (0.02–0.04)	0.02 (0.02–0.03)

* CrI: Credible interval.

## Data Availability

The data used in the analyses can be completely obtained from the corresponding author by reasonable request.

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
