# Peer review of "Modelling the Spatial Distribution of ASF-Positive Wild Boar Carcasses in South Korea Using 2019–2020 National Surveillance Data"

_animals, 2021, doi:10.3390/ani11051208_

Round 1
Reviewer 1 Report
The authors present very important work that could be very valuable to refine surveillance systems for the detection of ASF in wild boar.
9-20 - Revise the English of the “Simple Summary”
Line 9 – Delete “virus”, which is written twice
30 – Important to mention the dimensions of these hexagons
50 – Worth mentioning the genotype (II)
54 – Sus scrofa in Italics
59 – I would specify that these scavengers refer to wild boar. I’d assume other scavenging animals will have little to now influence on ASF persistence and spread
108-109 – It would be good to present the numbers of positives found through passive surveillance, active search, hunted and trapped. Results from Europe show low percentages of positives (and therefore low cost-effectivity of the system) of hunted wild boar. It would be valuable information to add.
111-12 – Type of specimen is carcass or captured. In which category do hunted wild boar fall?
113-114 – It should be explained somewhere the reasoning why trapped animals were excluded from the analysis.
113-14 – Only carcasses were considered for the study, but in what category are hunted animals? Under “carcass” or under “captured”? From an epidemiology point of view, they should fall in the same category as trapped animals. However it is not clear from the description. If also hunted animals were used for the analysis, it raises an issue. Experience from Europe shows that positive hunted animals (unless they were sick, i.e. hunted through sanitary culling), would be hunted during the incubation period, before they showed clinical signs. Otherwise, sick wild boar tend to go into hiding and rest when they start showing signs, which makes it difficult to hunt them. Based on this, one could argue that the factors behind the finding of carcasses will vary on how these positive animals were found. For example, passive surveillance (which depends on human density) will only lead to finding of dead carcasses, not hunted ones. If hunted animals have been included, it is necessary for the authors to describe how they accounted for the type of positive animal (hunted vs. found dead).
370 – The sentence says Figure 3E refers to first period, but in the Figure, it is listed within the second period. Shouldn’t it be 3A?
378 – I believe this is wrongly labelled as figure 4 (shouldn’t it be figure 3?)
411 – I was under the impression that not only carcasses but also hunted and trapped animals were used in the analysis.
429 – What do you mean by introduction through scavengers and other mechanical routes?
533 - Some more limitation of the study worth mentioning are 1) the fact that the performance of the model was not validated with new wild boar cases; and 2) the fact that carcasses are found where they were searched, meaning that search efforts where not homogeneous all over the study area, i.e. there are likely many carcasses that were never found.
It would be good if the authors could extract some recommendations from the findings as to how surveillance systems, in general, but also specifically for South Korea, could be improved.
Author Response
Dear Reviewer 1.
Thank you for your time and productive comments.
Please refer to the attached file.
Best regards,
Jun-Sik Lim, Timothee Vergne, Son-Il Pak, and Eutteum Kim

Reviewer 2 Report
The manuscript of Lim et al. describes a spatial zero-inflated Poisson regression model to identify risk factors associated with the presence of ASF positive wild boar carcasses in the Republic of Korea. While investigating mainly ecological indices based on remote sensing data, some potential risk factors were not included due to unavailability in the study region (military restrictions at the border to North Korea). In principle, the study was described in a comprehensive way. However, improvements were suggested to provide a sound background for the study. In parts of the manuscript a language check is needed. Please find my detailed comments in the attached file.

Author Response
Dear Reviewer 2.
Thank you for your time and reproductive comments.
Please refer to the attached file.
Best regards,
Jun-Sik Lim, Timothee Vergne, Son-Il Pak, and Eutteum Kim
